# EXPLORATION IN REINFORCEMENT LEARNING WITH DEEP COVERING OPTIONS

**Yuu Jinnai**
Brown University

**Jee Won Park**
Brown University

**Marlos C. Machado**
Google Brain

**George Konidaris**
Brown University

## ABSTRACT

While many option discovery methods have been proposed to accelerate exploration in reinforcement learning, they are often heuristic. Recently, covering options was proposed to discover a set of options that provably reduce the upper bound of the environment's cover time, a measure of the difficulty of exploration. However, they are constrained to tabular tasks and are not applicable to tasks with large or continuous state-spaces. We introduce *deep covering options*, an online method that extends covering options to large state spaces, automatically discovering task-agnostic options that encourage exploration. We evaluate our method in several challenging sparse-reward domains and we show that our approach identifies less explored regions of the state-space and successfully generates options to visit these regions, substantially improving both the exploration and the total accumulated reward.

## 1 INTRODUCTION

Temporal abstraction, often formalized via the options framework (Sutton et al., 1999), has the potential to greatly improve the performance of reinforcement learning (RL) agents by representing actions at different time scales. However, the question of which options an agent should construct, and the related question of what objective function that option construction process should be optimizing, remain open. One recent approach is to construct options that aid exploration by providing agents with more decisive behavior than the dithering common to random exploration (e.g., Menache et al., 2002; Stolle and Precup, 2002; Şimşek and Barto, 2004; Şimşek et al., 2005; Şimşek and Barto, 2009; Machado et al., 2017; Eysenbach et al., 2019). The Laplacian (Chung, 1996), the matrix extracted from the graph induced by the agent's policy and the dynamics of the environment, is often used when discovering options for exploration (e.g., Machado and Bowling, 2016; Machado et al., 2017; 2018; Jinnai et al., 2019b). The options discovered with such an approach encourage agents to navigate to parts of the state space that are infrequently visited. However, the existing methods either lack a principled way of constraining the number of discovered options (e.g., Machado and Bowling, 2016; Machado et al., 2017; 2018) or are limited to the tabular setting (e.g., Jinnai et al., 2019b).

In this paper we show how recent developments in eigenfunction estimation of the Laplacian (Wu et al., 2019) can be used to extend a principled approach for option discovery (Jinnai et al., 2019b) to the non-linear function approximation case. This new algorithm for option discovery, *deep covering options*, is computationally tractable and it is applicable to environments with large (or continuous) state-spaces. Despite methods that learn representations generally being more flexible, more scalable, and often leading to better performance, before this paper, covering options could not be easily combined with modern representation learning techniques. Deep covering options discovers a small set of options that encourage exploration by minimizing the agent's *expected cover time*—the expected number of steps required to visit every state in the environment (Broder and Karlin, 1989). Moreover, unlike most previous approaches to discovering options for exploration, it can be applied to both settings where a pretraining (unsupervised) phase is available (e.g., Eysenbach et al., 2019) and to the traditional, fully online, setting.

We evaluate our method, in both settings, in three different platforms to demonstrate its applicability in a wide range of domains. First, we apply it to the Pinball domain (Konidaris and Barto, 2009), which has a discrete action-space and a continuous state-space. Second, we apply it to three MuJoCo control tasks (Todorov et al., 2012), which are continuous state- and action-space domains. In all of

these domains, our method improves over the baseline. Finally, we perform a qualitative analysis of our method in three Atari 2600 games (Bellemare et al., 2013) to demonstrate its potential in domains with very large state-spaces. Deep covering options successfully finds under-explored regions of the state space and builds options to target those regions.

## 2 BACKGROUND AND RELATED WORK

We assume the standard reinforcement learning setting (Sutton and Barto, 1998), where the environment is modeled as a Markov Decision Process (MDP), $(\mathcal{S}, \mathcal{A}, T, R, \gamma)$, where $\mathcal{S}$ is the set of states, $\mathcal{A}$ is the set of actions, $T : \mathcal{S} \times \mathcal{A} \times \mathcal{S} \to [0, 1]$ is the state transition function, $R : \mathcal{S} \times \mathcal{A} \to \mathbb{R}$ is the reward function, and $0 \leq \gamma \leq 1$ is the discount factor.

We use the *options framework* (Sutton et al., 1999) to represent temporally extended actions. It defines an option as a triple $(\mathcal{I}, \pi, \beta)$, where $\mathcal{I} \subseteq \mathcal{S}$ is the set of states in which the option can initiate, $\pi : \mathcal{S} \to \Pr(\mathcal{A})$ is the policy the agent follows when that option is being executed, and $\beta : \mathcal{S} \to [0, 1]$, is the termination condition. We refer to a set of states in which $\beta(s) = 1$ as a termination set.

### 2.1 RELATED WORK

Many option discovery algorithms are based on the reward signals generated by the environment and are thus task dependent. These methods often decompose the trajectories reaching the rewarding states into options. Several papers have proposed generating options from trajectories reaching these rewarding states (e.g., McGovern and Barto, 2001; Menache et al., 2002; Konidaris and Barto, 2009), while other approaches use the observed rewards to generate options with gradient descent (e.g., Mankowitz et al., 2016; Bacon et al., 2017; Harb et al., 2018; Tiwari and Thomas, 2019). These approaches are often ineffective in sparse reward problems, where only a few state-action pairs lead to a positive reward.

Fewer papers have tackled the problem of option discovery for exploration without using reward signals. Eysenbach et al. (2019) proposed to generate options maximizing an information theoretic objective so that each option generates diverse behavior. While many option discovery methods are limited to discrete state and action space tasks, their method can generate options that solve many continuous control tasks, even when ignoring the environment's reward function. Machado et al.; Machado et al. (2017; 2018) proposed eigenoptions, a method to generate options using the Laplacian eigenvectors (Chung, 1996). Their approach is similar to covering options but requires the set of options to be orthogonal to each other and introduces a prohibitively large number of options at each iteration. Several papers have proposed identifying subgoal states without reward information through graph concepts such as clustering (Menache et al., 2002; Şimşek et al., 2005), visitation statistics (Şimşek and Barto, 2004; Stolle and Precup, 2002), and betweenness centrality (Şimşek and Barto, 2009). As they use graph algorithms to discover subgoals, their scope is often limited to tabular domains.

### 2.2 COVERING OPTIONS

Covering options (Jinnai et al., 2019b) is an approach that minimizes the *expected cover time* of a uniformly random policy by augmenting the agent's action set with options obtained from the eigenvector associated with the second smallest eigenvalue of the Laplacian. Covering options can be seen as increasing the likelihood that a random walk is going to lead to a rewarding state since the expected cover time is the time required for a random walk to visit all the vertices in a graph (Broder and Karlin, 1989). Covering options achieves such an objective by minimizing the upper bound of the expected cover time, $\mathbb{E}[C(G)]$, which is given by the second smallest eigenvalue of the normalized Laplacian, $\lambda_2$, also known as the *algebraic connectivity* (Fiedler, 1973):

$$\mathbb{E}[C(G)] \leq \frac{n^2 \ln n}{\lambda_2}(1 + o(1)), \tag{1}$$

where $n$ is the number of vertices of the graph. Equation 1 shows that the larger the algebraic connectivity, the smaller the upper bound of the expected cover time.

Intuitively, algebraic connectivity represents how densely the graph is connected. The eigenvector $f$ corresponding to $\lambda_2$ is an embedding of a graph to a one-dimensional interval where nodes connected

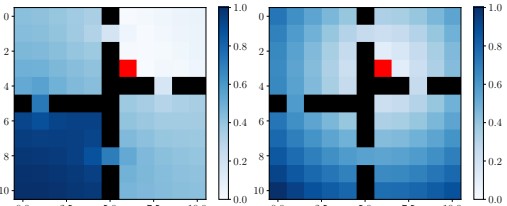

Figure 1: The distance between the red state and all other states, measured via the second eigenvector (left) and Euclidean distance (right). The second eigenvector captures the connectivity of the graph, so distances reflect path lengths in the graph; the pair of nodes with the maximum and minimum values are the farthest apart. Figure is adapted from Jinnai et al. (2019b), Figure 2.

Figure 2: Comparison between options generated by deep covering options (left) and covering options (right). Blue regions represent states in the initiation set and shaded regions states in the termination set. Generated options have initiation and termination sets consisting of a single state, making them impractical in large state-spaces.

by an edge tend to be placed nearby (see Figure 1, adapted from Jinnai et al., 2019b). A pair of nodes with the maximum and minimum value in $f$ are the most distant nodes in the embedding space. Connecting these two nodes greedily maximizes the algebraic connectivity to a first order approximation (Ghosh and Boyd, 2006). Covering options works as follows:

1. Compute the second smallest eigenvalue and the corresponding eigenvector $f$ of the Laplacian exactly by solving the following constraint optimization problem:

$$\lambda_2 = \inf_{\substack{f^T A\mathbf{1}=0 \\ f^T Af=1}} G(f) \qquad G(f) = \frac{1}{2} \sum_{s \in \mathcal{S}} \Big[ \big(f(s) - f(s')\big)^2 A(s,s') \Big], \qquad (2)$$

where $A$ is the adjacency matrix of the state-space graph where the entry at $(s, s')$ is 1 if $s$ and $s'$ are adjacent and 0 otherwise.

2. Let $v_i$ and $v_j$ be the state with largest and smallest value in the eigenvector respectively. Generate two options; one with $\mathcal{I} = \{v_i\}$ and $\beta = \{v_j\}$ and the other one with $\mathcal{I} = \{v_j\}$ and $\beta = \{v_i\}$. Each option policy is the optimal path from the initial state to the termination state.

3. Set $G \leftarrow G \cup \{(v_i, v_j)\}$ and repeat the process until the number of options reaches a threshold.

While this method is an efficient algorithm with performance guarantees, it is limited to small discrete MDPs as it requires a state-space graph. Moreover, explicitly computing the matrix that encodes the environment's adjacency matrix is unrealistic beyond small problems. Finally, the method is constrained to *point options* where both the initiation and termination sets consist of a single state (Jinnai et al., 2019a). Options generated by this method are therefore only executable at a single state. This is not useful for tasks with large (or continuous) state-spaces as the probability of visiting the state in the initiation set of the option tends to zero. Even if the agent visits the state in the option's initiation set and starts following the corresponding option's policy, the probability of reaching the state in the termination set is also small (see Figure 2). In the next section we introduce an approach that addresses these limitations.

## 3 DEEP COVERING OPTIONS

We propose *deep covering options*, a new algorithm that finds options that speed-up exploration in domains with large (or continuous) state-spaces. It directly seeks to optimize an objective for exploration. If the objective function is optimized, the options generated by the algorithm greedily maximize the algebraic connectivity of the underlying state-space graph to a first order approximation (Ghosh and Boyd, 2006), which in turn minimize the upper bound on the expected cover time. Deep covering options consists of four steps (see Algorithm 1):

---

**Algorithm 1** Deep covering options

---

1: Input: Set of state-transitions $\mathcal{H}$, a percentile $0 \leq k \leq 100$
2: Compute $f$ by minimizing $\tilde{G}(f)$ using $\mathcal{H}$ (Equation 5)
3: $\beta' \leftarrow k$-th percentile value of $f$ in $\mathcal{H}$
4: $\beta_o(s) \leftarrow \begin{cases} 1 & \text{if } f(s) < \beta' \\ 0 & \text{otherwise} \end{cases}$
5: $\mathcal{I}_o \leftarrow \{s|\beta_o(s) = 0, s \in S\}$
6: Train $\pi_o$ off-policy by maximizing the total accumulated pseudo-rewards $r_o = f(s) - f(s')$ using $\mathcal{H}$ and $f$
7: Return $(\mathcal{I}_o, \pi_o, \beta_o)$

---

1. compute an eigenfunction of the Laplacian of the state-space graph approximately (line 2 in Algorithm 1),

2. identify an under-explored region in the state-space using the eigenfunctions (line 3),

3. set the under-explored region as the termination set and set the compliment of it as the initiation set (line 4, 5),

4. train a policy of the option using the pseudo-reward induced by the eigenfunctions (line 6).

There are two problems in Equation 2 that prevent its applicability to non-tabular domains. First, the equation requires the adjacency matrix $A$ as input. Second, a constrained optimization problem is hard to solve using gradient-based methods. We address these issues by approximating the computation of the Laplacian with the following objective (Wu et al., 2019, Equation 6):

$$\tilde{G}(f_1, f_2, ..., f_d) = \frac{1}{2}\mathbb{E}_{(s,s')\sim\mathcal{H}}\Big[\sum_{k=1}^{d}\big(f_k(s) - f_k(s')\big)^2\Big]$$
$$+ \eta\mathbb{E}_{s\sim\rho,s'\sim\rho}\Big[\sum_{j,k}\big(f_j(s)f_k(s) - \delta_{jk}\big)\big(f_j(s')f_k(s') - \delta_{jk}\big)\Big], \qquad (3)$$

where $\mathcal{H}$ is the set of sampled state-transitions, $\rho$ is a distribution of states in the dataset ($\rho(s)$ is the number of occurrence of $s$ in $\mathcal{H}$ divided by the size of $\mathcal{H}$), $\eta$ is the Lagrange multiplier, and $\delta_{jk}$ is 1 if $j \neq k$ and 0 otherwise. Such an expression, inspired by spectral graph drawing theory, uses the repulsive term (the summation multiplied by $\eta$) to ensure the functions $f_1, ..., f_d$ are orthogonal to each other. Unlike $G$, $\tilde{G}$ is a constraint-free objective to compute the eigenfunction, only requiring trajectories instead of the state-space graph.

As we only require the second eigenfunction (unlike eigenoptions), we can simplify the objective function to take only two arguments:

$$\tilde{G}(f_1, f_2) = G(f_1, f_2) + \eta\mathbb{E}_{s\sim\rho,s'\sim\rho}\Big[\sum_{j,k}\big(f_j(s)f_k(s) - \delta_{jk}\big)\big(f_j(s')f_k(s') - \delta_{jk}\big)\Big]. \qquad (4)$$

Assume $G(f_1) \leq G(f_2)$ without loss of generality. $G(f_1) = 0$ and $f_1$ is a constant function because the first eigenvalue of the Laplacian matrix is zero. To simplify the equation, we assume $f_1 = \mathbf{1}$ without loss of generality. Then:

$$\tilde{G}(f) = \tilde{G}(\mathbf{1}, f) = \frac{1}{2}\mathbb{E}_{(s,s')\sim\mathcal{H}}\Big[\big(f(s)-f(s')\big)^2\Big]+\eta\mathbb{E}_{s\sim\rho,s'\sim\rho}\Big[\big(f(s)^2-1\big)\big(f(s')^2-1\big)+f(s)^2f(s')^2\Big].$$
$$(5)$$

Deep covering options compute the second eigenfunction $f$ by minimizing $\tilde{G}(f)$ instead of $G(f)$ (see Algorithm 1). Our objective function only needs sampled state-transitions $\mathcal{H}$ instead of a complete state-space graph. As it is an unconstrained optimization problem, we can optimize by simple gradient-based methods. The objective function is essentially the same as the objective function of covering options which has theoretical guarantee on the expected cover time but computed approximately so that it scales to large or infinite state-space domains.

| $k$ | **Reward** |
|----|----------|
| 5 | $2.5311 \pm 1.71$ |
| 10 | $3.1873 \pm 3.33$ |
| 30 | $3.2210 \pm 4.54$ |
| 50 | $3.0748 \pm 3.28$ |

Table 1: The effect of the size of the termination set (percentile $k$) on the performance of Deep covering options in Pinball with 3 options. Reward is averaged over 100 episodes and 5 runs.

While covering options is constrained to options with the initiation set consisting of a single state, we set the termination set as a set of states with $f$ value smaller than its $k$-th percentile. As proposed by Machado et al. (2017; 2018), we define the initiation set to be the complement of the termination set. We train the option policy off-policy, maximizing the total pseudo-reward $r_o = f(s) - f(s')$ so that it learns to reach the termination set (i.e., the set of states with $f(s) < \beta'$).

## 4 EXPERIMENTS

We evaluate our method in both the online setting and the setting in which a pretraining phase is available. We use three different platforms: the Pinball domain (Konidaris and Barto, 2009), three MuJoCo control tasks (Todorov et al., 2012), and three Atari games (Bellemare et al., 2013). See the Appendix for the experimental details.

### 4.1 OFFLINE OPTION DISCOVERY

We first consider the setting in which the agent collects samples in the environment for a given number of time steps before being given a reward signal to maximize.

**Pinball** In the Pinball domain the goal is to maneuver a small ball from a start state to a goal state (Figure 3a; Konidaris and Barto, 2009). The state-space consists of four continuous variables, the coordinates of the ball position $(x, y)$ and the velocity $(\dot{x}, \dot{y})$. There are five primitive actions: incrementing or decrementing $\dot{x}$ or $\dot{y}$ by a small amount or leaving them unchanged. The ball bounces on colliding with obstacles. In order to reach the goal (red cross) from the initial position (purple circle), the agent must get through one of the narrow passages while taking the bounce into consideration. The agent receives a reward of 10 upon arrival at the goal and of -0.001 in each other time step. The start state is fixed throughout the training.

To generate an option, we sampled 100 trajectories of 1000 time steps in which the agent selects between the available actions and options uniformly at random. We trained a neural network to learn the eigenfunction by minimizing $\tilde{G}$ using the sampled state-transitions (see Equation 5). We evaluated with the threshold percentile $k = \{5, 10, 30, 50\}$ and selected 30 as it performed the best (Table 1). We used Q-learning (Watkins and Dayan, 1992) with Fourier basis linear function approximation (Konidaris et al., 2011) to train the option policy off-policy using the sampled trajectories but using the pseudo-reward $r_o$ (see Algorithm 1). We repeat this process with the generated option added to the agent's action set.

We evaluate the performance of the agent with access to the discovered options to evaluate the claim that these options do indeed allow the agent to collect more rewards by making it more capable of navigating through the state space. The agent has access to both these computed options and primitive actions, and uses Q-learning with the Fourier basis to train the high-level policy.

Figure 3e depicts the agent's performance with a varying number of options. The proposed algorithm significantly outperforms the flat baseline. We also evaluated the performance of flat Q-learning pretrained with reward signals for the same number of episodes the hierarchical agents were (base-pretrained). While the options are generated without reward information, the performance of the agent with the option set is close to the performance of the agent trained with reward information, showing that such an approach does not hinder performance even in a single task setting.

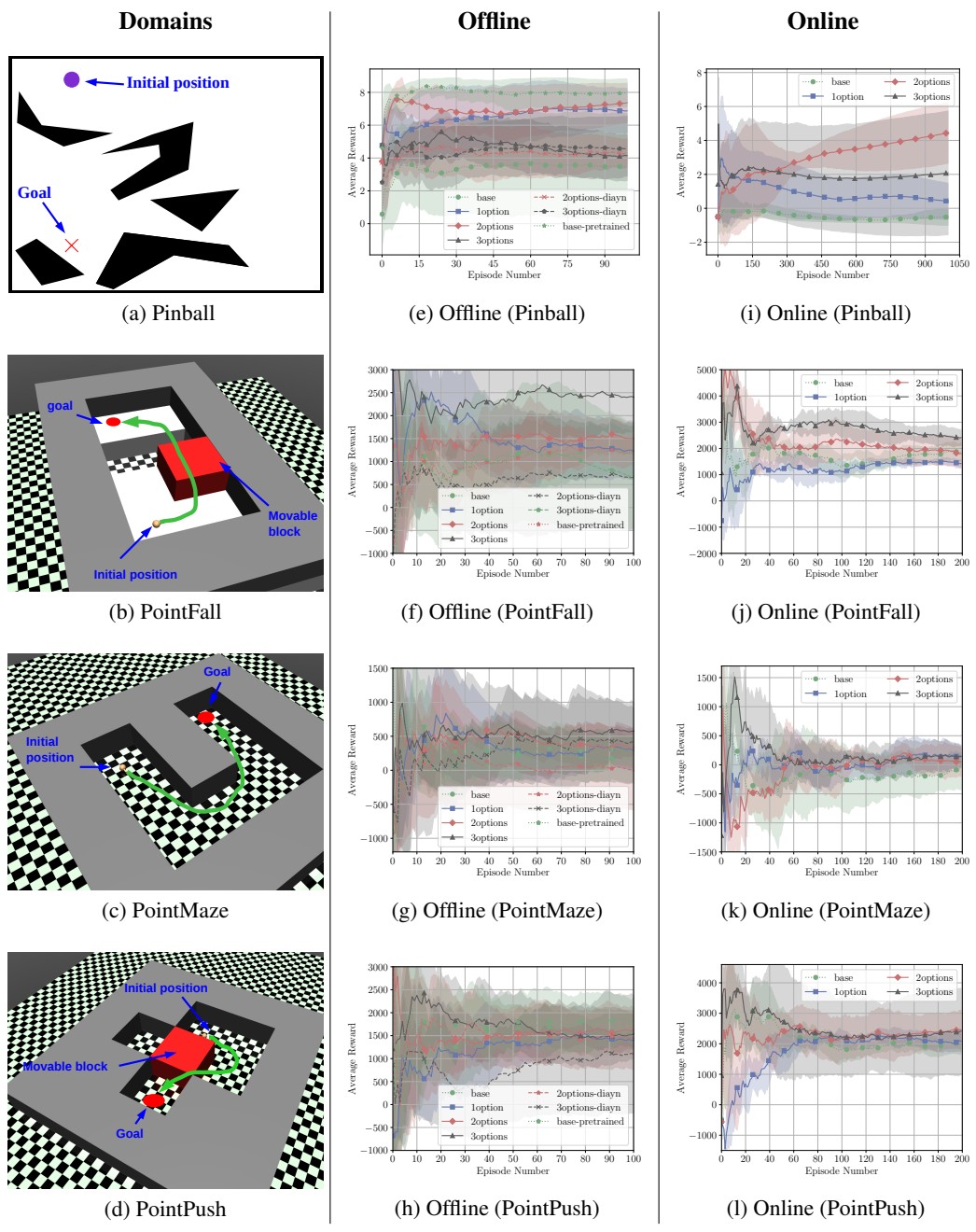

Figure 3: Performance of online option discovery agents, averaged over 5 runs. The shaded area shows the standard deviation. In PointFall (Figure 3b), the agent must push the movable block into a chasm to make a bridge that allows it to reach the goal. In PointMaze (Figure 3c), the agent must first move away from the goal (in terms of L2 distance) to successfully reach it, since the corridor is U-shaped. The green arrow shows successful trajectories. In PointPush (Figure 3d) a greedy agent would move forward and push the movable block into the path to reach the goal. To reach the goal, it must push a movable block to the right to clear the path towards the goal.

The termination set generated by deep covering options tend to be larger than options which seek to minimize the size of the termination set (e.g. Harutyunyan et al., 2019). The results indicate that interpretable options are not necessarily efficient for reducing cover time. This is a known behavior of option-discovery algorithms based on spectral methods such as eigenoptions (Machado et al., 2017).

We also compared our approach to Diversity Is All You Need (DIAYN) (Eysenbach et al., 2019). Like our method, DIAYN was recently proposed to generate exploratory options without using reward signals. While many option discovery methods are limited to discrete state-space tasks, DIAYN can operate in continuous control tasks. We trained DIAYN for the same length of pretraining steps (300 episodes each one being 1000 steps long) to generate a set of options. See the appendix for details of the agent. While Eysenbach et al. (2019, Section 5.1) assumed that one can pick the option (i.e. skill) with highest reward for the task and trained the agent starting from that single option, we use the more realistic assumption that the agent has no prior information of which option is most useful for a given task. Based on this assumption, we evaluated an agent equipped with all the generated options and the primitive actions so that the agent must learn which option is most useful for the given task by itself. We used Q-learning with Fourier basis to train the high-level policy. As the termination condition of DIAYN is not defined by Eysenbach et al. (2019), we tested the termination probability of 0.0, 0.01, 0.1, and 0.5 for any state, and picked 0.1 as it performed the best. We set the initiation set to be the whole state space, and evaluated the performance of DIAYN for up to three options. DIAYN outperforms the baseline that consists of only primitive actions. While DIAYN generates a diverse set of options by maximizing the mutual information between states and options, it does not consider state connectivity. As our algorithm takes into account the connectivity of the states to generate diverse set of options, it successfully finds an option which leads to a state close (in terms of the number of steps to reach) to the goal state with high probability, resulting in better performance than DIAYN.

**MuJoCo** Next, we evaluated our method in three simulated continuous control tasks introduced by Nachum et al. (2018): PointFall, PointMaze, and PointPush. The agent receives a reward signal of value 8000 when it reaches the goal and a reward signal of value $-$(L2 distance to the goal)/(maximum possible L2 distance to the goal) otherwise. The agent cannot reach to goal state just by maximizing the immediate reward given by the L2 distance to the goal (see Figure 3). PointFall is difficult for a plain agent because if it just follows the immediate reward it falls off the cliff and can never reach the goal whereas in PointMaze and PointPush are relatively easy for a plain agent as it can eventually reach the goal. The start state is fixed to the same position throughout the training but the initial rotation of the agent is set randomly. We sampled 200 episodes of length 2000 with a uniform random policy to generate each option. We trained the option's policy using deep deterministic policy gradient (DDPG Lillicrap et al., 2016), used the same hyperparameters as Wu et al. (2019) for DDPG and for learning the eigenfunction $f$ (Wu et al., 2019, Appendix D2.2), and set the threshold percentile $k = 10$.

The high-level policy chooses options with Double Deep Q-learning (van Hasselt et al., 2016). We train the agents for 100 episodes, each 2000 time steps long. Figure 3f, 3g, and 3h show the performance with varying number of options. While the performance improvement is small in PointPush and PointMaze, where even a flat agent can easily reach the goal, it is significantly improved in PointFall, which is hard to solve without an efficient exploration strategy. Our method sometimes even outperforms the agent pretrained with reward signal available (base-pretrained). We trained DIAYN for continuous control tasks too, but it did not outperform the baseline in PointFall, PointMaze, and PointPush. See the Appendix for experimental details.

## 4.2  Online Option Generation

In the previous section we evaluated option discovery methods assuming that the agent can collect samples by interacting with the environment prior to solving the task itself. We now evaluate the proposed algorithm in the online setting, where the agent generates options using trajectories sampled during the learning phase. At the beginning of training the agent only has access to primitive actions; it then generates one new option every 2000 time steps using the observed data until the number of options reaches a pre-specified threshold. The option policy is trained off-policy using the trajectories sampled while training. Thus, our method does not require any extra samples. We use the same learning algorithms and hyperparameters as the offline experiments. These experiments aim to evaluate whether the cost of learning the options when a task is given is prohibitive.

We train the agents for 1000 episodes, each 500 time steps long for Pinball. In the continuous control tasks we train the agents for 200 episodes, each 1000 time steps long. Figure 3i, 3j, 3k, and 3l (right most column) depict the agent's performance with a varying maximum number of options. Overall, the proposed algorithm significantly improved performance compared to the baseline in most tasks (i.e., Pinball, PointMaze, and PointFall). As in the previous section, we did not see a major improvement in PointPush where the agent can easily discover near-optimal policies using only primitive actions. These results suggest that the proposed method is not only useful for pretraining, but can also discover useful options during training and successfully speed up learning without additional samples.

### 4.3 QUALITATIVE EVALUATION

We now show how the options discovered in the pretraining phase (see Section 4.1) improve the agent's exploration capabilities. Figure 4a depicts the $(x, y)$ positions visited by the agent in the 10 trajectories generated by a random walk when using only primitive actions. Note that the agent rarely gets through the corridor. Figure 4b visualizes the termination set and one of the trajectories generated by the first option discovered by the algorithm. The shaded region indicates the option termination set. Notice that the algorithm successfully discovers the region under-explored by the agent (Figure 4a). Figure 4c shows the $(x, y)$ positions of the states visited by 10 trajectories in total, with 5 trajectories only using primitive actions and 5 trajectories with the first option available to the agent. The agent now consistently gets through the narrow passages. Figure 4d shows one of the trajectories generated by the second discovered option. The same process can be observed when a second and third options are added to the action set. They keep identifying under-explored regions of the state space and further narrow down the termination set to visit these regions (Figures 4d). These results suggest that the proposed method successfully extends the frontier of the exploration by discovering options incrementally.

The same intuition holds for continuous control tasks. This can be seen in Figures 4e and 4f. Figure 4e shows the states visited by 10 trajectories generated by a random walk using primitive actions in PointMaze; the agent does not deviate far from its start state. Figure 4f shows the state visited by 10 trajectories in total, 5 trajectories with primitive actions and 5 with the first option available to the agent; the agent is now able to explore further along the corridor. By incrementally discovering options our method generates options to navigate through the maze without any reward information. This becomes evident in Figure 4g, which depicts the trajectory followed by the first (green), second (yellow), and third (purple) options. Each option explores more deeply into the state space.

To demonstrate the potential of the proposed method in a domain with a very large discrete state-space, we visualize the termination set of options generated in three Atari games from the Arcade Learning Environment (Bellemare et al., 2013): Montezuma's Revenge, MsPacman, and Amidar. See the Appendix for the experimental details. The figures suggest that the options aim to visit different regions of the state space, promoting exploration in these games as well. Importantly, unlike other approaches evaluated in Atari games (e.g., Machado et al., 2017; 2018), the options our method generates need not to be curated by an expert, who filters out non-meaningful options. Nevertheless, further analyses down the eigenspectrum are required for a better understanding of the diversity and utility of the discovered options.

## 5 CONCLUSION

Deep covering options is a new method for learning options to explore the state-space efficiently in a task-agnostic way. By minimizing expected cover time, it automatically discovers less-explored regions of the state-space and generates options to reach those regions. Our algorithm is inspired by strong theoretical results in the tabular case while being computationally practical in large domains. We demonstrated the use of our method in a pretraining setting as well as for the traditional online setting. In pretraining experiments we showed that the method is able to generate task-agnostic options which expand the frontier of the known regions of the state space without any reward information and successfully improves the performance of the agent in continuous control tasks. In online experiments we showed that the proposed method can also discover useful options during training and successfully speeds up learning without additional sampling.

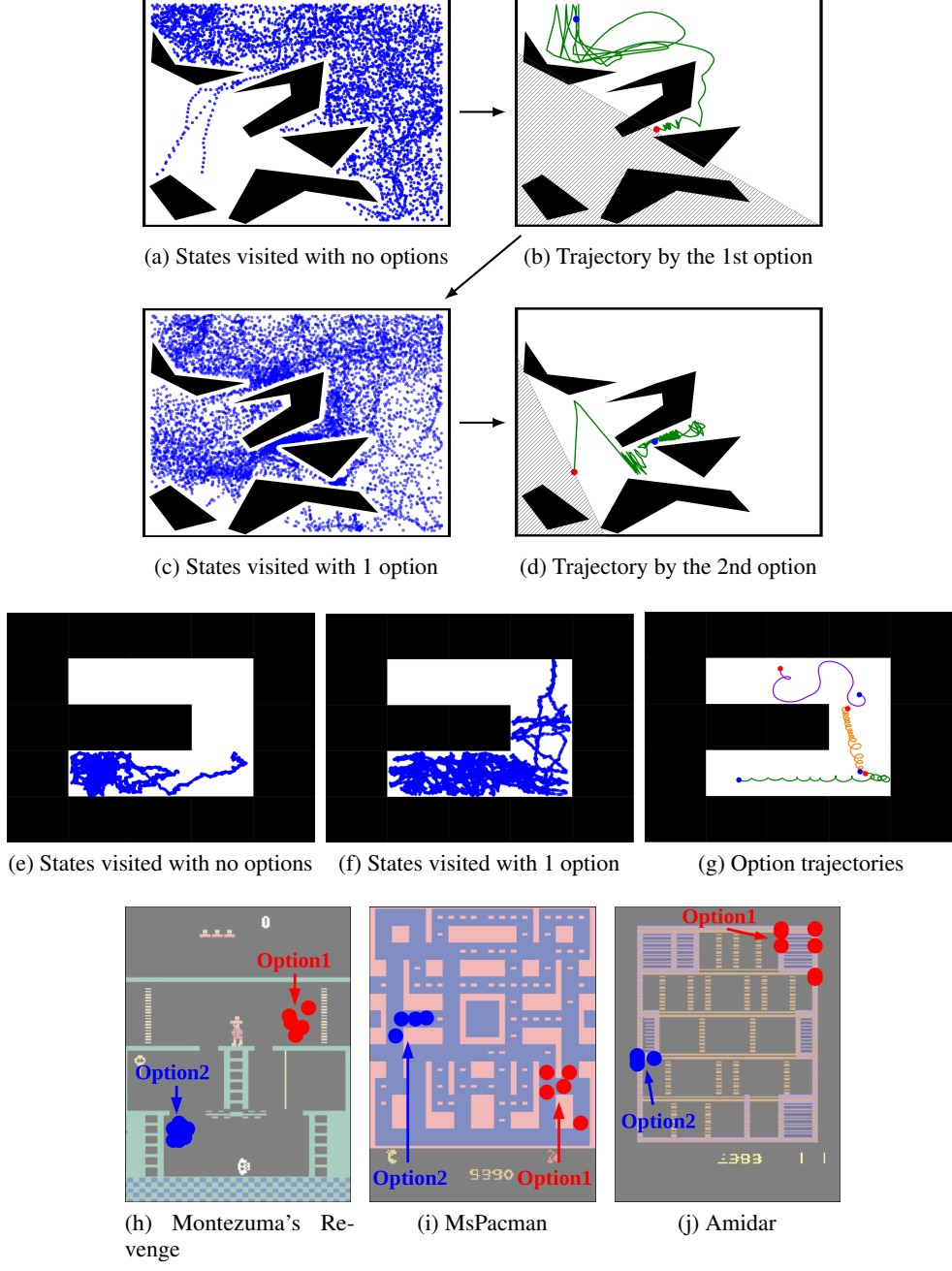

Figure 4: Options generated by offline option discovery. (a, c, e, f) States visited by a random walk without and with options. (b, d) Trajectories obtained by the generated options. Shadowed regions in the figures approximately show the $(x, y)$ coordinate of the termination set when the velocity is 0. The ball may not terminate in the shaded region for velocity higher than 0. (g) Trajectories by the first, second, and third option in PointMaze. (h–j) Termination set of the options in Atari games.

ACKNOWLEDGMENTS

This research was supported in part by DARPA under agreement number W911NF1820268, AFOSR Young Investigator Grant agreement number FA9550-17-1-0124.and the ONR under the PERISCOPE MURI Contract N00014-17-1-2699. The U.S. Government is authorized to reproduce and distribute reprints for Governmental purposes notwithstanding any copyright notation thereon. The content is solely the responsibility of the authors and does not necessarily represent the official views of DARPA, the ONR, or the AFOSR.

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

# A EXPERIMENTAL SETUP

## A.1 PINBALL

We used Q-learning ($\alpha = 0.1, \gamma = 0.99, \epsilon = 0.05$) with linear function approximation with a 3rd-order Fourier basis Konidaris et al. (2011) to train the policy of the option off-policy using the sampled trajectories but using the intrinsic reward $r_o$ (Algorithm 1). We set the percentile to $k = 30$. We set the Lagrange multiplier $\eta$ to 1.0.

We used Q-learning ($\alpha = 0.1, \gamma = 0.99, \epsilon = 0.05$) with a 3rd-order Fourier basis to train the high-level policy. The option's policy is obtained with Q-learning ($\alpha = 0.1, \gamma = 0.99, \epsilon = 0.05$) with linear function approximation with a 3rd-order Fourier basis. We used the same setup for both offline and online option discovery.

### A.1.1 DIAYN

The actor and the critic are implemented with 3 hidden layers with 256 units each followed by a ReLU activation function with Adam optimizer with a step size of 0.005. We used neural networks for the actor and the critic as it outperformed an agent with the actor and the critic implemented by a linear approximator using 3rd order Fourier basis Konidaris et al. (2011). Our discriminator network consists of 2 hidden layers with 256 units each followed by a ReLU activation function. We trained the discriminator with Adam optimizer with a step size of 0.001.

We used Q-learning ($\alpha = 0.1, \gamma = 0.99, \epsilon = 0.05$) with linear function approximation with 3rd-order Fourier basis to train the high-level policy. As the termination condition of DIAYN is not defined in the work by Eysenbach et al. (2019), we tested the termination probability of 0.0, 0.01, and 0.1, and 0.5 for any states. We picked 0.1 as it performed the best.

## A.2 MUJOCO

We trained the option's policy using deep deterministic policy gradient Lillicrap et al. (2016). We used the same hyperparameters as Wu et al. (2019) for DDPG and the eigenfunction $f$ (see Appendix D.2.2 in Wu et al. (2019)). We set the threshold percentile $k = 10$. We set the Lagrange multiplier $\eta$ to 1.0.

The high-level policy chooses options with double deep Q-learning van Hasselt et al. (2016). The Q-network consists of two fully connected layers with 400 units with a batch normalization and a ReLU in between. We trained it with the Adam optimizer using a step size of 0.0001 and a batch size of 64. We updated a target policy every step by update rate of 0.001. We set $\epsilon$ to 0.05. We used the same setup for both offline and online option discovery.

We evaluated DIAYN for the continuous control tasks. The actor and the critic consist of three hidden layers with 256 units each followed by a ReLU activation function with Adam optimizer with a step size of 0.005. Our discriminator network consists of two hidden layers with 256 units each followed by a ReLU activation function. We evaluated Adam optimizer with a step size of 0.001 and 0.005 for the discriminator. We used double DQN van Hasselt et al. (2016) to implement the high-level policy. The Q-network consists of three fully connected layers with 400 units with a batch normalization and a ReLU in between. We also evaluated the Q-network with two fully connected layers. We trained it with the Adam optimizer with a step size of 0.001 and a batch size of 64. We updated a target policy every step by update rate of 0.001. We also tried update rate of 0.01. For the setup we have tried, DIAYN did not outperform the baseline in all PointMaze, PointPush, and PointFall.

## A.3 ARCADE LEARNING ENVIRONMENT

We use the screen image as the observation and train $f$ using the sampled pixel images. We use a transformation from the original (210, 160, 3) dimensional RGB image to a (105, 80) grayscale image. We set the threshold percentile $k = 4$. We set the Lagrange multiplier $\eta$ to 1.0. We sampled 200 trajectories each being 2000 steps long. The eigenfunction is learned with a convolutional neural network with 2 convolution layers (32 8x8 filters with stride 4 and 64 4x4 filters with stride 2) and a fully-connected hidden layer with 400 units and ReLU in between. We trained the policy of the option with double deep Q-learning van Hasselt et al. (2016). The Q-network consists of 2 convolution

layers (32 8x8 filters with stride 4 and 64 4x4 filters with stride 2) and a fully-connected hidden layer with 400 units and ReLU in between. We trained it with Adam optimizer with a step size of 0.0001 and a batch size of 64. We updated a target policy every step by update rate of 0.001. We set $\epsilon$ to 0.05. We sampled 50 trajectories and plotted the positions of the player agent when the option terminated. The plots depict the termination set of the generated options. The figures suggest that different options aim at visiting different regions of the state space, promoting exploration in these games as well. Importantly, differently from other approaches evaluated in Atari games (e.g., Machado et al., 2017; 2018), the options our method generates do not need to be curated by an expert, with the first eigenfunctions being able to generate meaningful options. Nevertheless, further analyses down the eigenspectrum are required for a better understanding of the diversity and utility of the discovered options.

