# OpenReview forum: "Exploration in Reinforcement Learning with Deep Covering Options"
_ICLR.cc/2020/Conference — Accept (Poster)_

### Official Review · AnonReviewer1 · 2019-10-20
**Official Blind Review #1**

**Rating:** 6

**Review:**

This paper proposes deep covering options. This method extends laplacian based option generation techniques to continuous domains. The resulting options are task independent and enable efficient exploration. The method can be applied in continuous state (and action) domains and options can be learnt in an online manner.

I favour acceptance of this paper. While the method seems to be mostly a combination of the ideas in 2 referenced previous works, the method seems to work well in a diverse set of circumstances.

Detailed comments:

-The proposed method seems mainly a combination of covering options with the techniques from (Wu et al.) to compute the eigenfunctions. As such it could be argued  to the core ideas in the paper aren’t very novel. Nonetheless, I found the approach interesting. Moreover, it is interesting to see an option generation technique that applies to such a wide range of problems.

-The paper gives a broad overview of related work and situates the method with respect to previous option generation methods. The background given on the laplacian, its eigenfunctions and their properties was short to the point of not being very clear. The algorithms are given without much context. I found myself referring to the earlier papers for a more thorough explanation of the core ideas. I would suggest shortening the other sections somewhat in favour of a more intuitive and self contained explanation.

- The effectiveness of the method is demonstrated on a set of very diverse problems that go well beyond the traditional grid worlds often used in option literature. The method is also compared to another unsupervised option generation method.

- It is interesting to see a method that can incrementally grow the set of options while learning, without any pre-training or requiring additional samples. This has the potential to have a large impact on large scale learning approaches

- What is the cost of repeatedly solving the minimization problem in (5)?

Minor comments:

- For several experiments the reward curve starts out high or rapidly becomes high and then decreases during learning. Can the authors explain this odd behaviour? This seems to mainly happen with the option methods in the continuous control domains

- Why are there no learning curves for the ATARI domain?


**Experience Assessment:**

I have published in this field for several years.

**Review Assessment: Checking Correctness Of Derivations And Theory:**

I assessed the sensibility of the derivations and theory.

**Review Assessment: Checking Correctness Of Experiments:**

I assessed the sensibility of the experiments.

**Review Assessment: Thoroughness In Paper Reading:**

I read the paper thoroughly.

---

> ### Author Response · Authors · 2019-11-12
> **Response to Reviewer #1**
>
> We appreciate the time you took reviewing our submission and hope that our response addresses some of your concerns.
>
>
> > The proposed method seems mainly a combination of covering options with the techniques from (Wu et al.) to compute the eigenfunctions. As such it could be argued  to the core ideas in the paper aren’t very novel.
>
> One of the main limitations of eigenoptions is that the existing algorithms discover too many options at once and it is not clear how to prune them. The work on covering options addressed this limitation but only in a constrained, theoretical setting. Extending this idea to the function approximation setting is actually really challenging. We argue that the strength of our paper is to show that covering options can be easily extensible to function approximator setting by applying and adapting the techniques shown by Wu et al.
>
>
> > What is the cost of repeatedly solving the minimization problem in (5)?
>
> The computation for solving the minimization problem was roughly the same as training the DQN agent for one iteration.
>
>
> > For several experiments the reward curve starts out high or rapidly becomes high and then decreases during learning. Can the authors explain this odd behaviour?
>
> We observed that the value of the options gets higher during the training and sometimes its value overly generalizes to states where calling the options are not appropriate (e.g. you cannot reach the terminal states easily under the policy of the option).
>
>
> > Why are there no learning curves for the ATARI domain?
>
> Finding efficient options to improve the exploration in the ATARI domain is very challenging. While Machado et al. (2017; 2018) showed that eigenoptions is applicable to the ATARI domain, they did not present learning curves. They constructed 1024 options from all the generated eigenfunctions and manually picked some of the options out of them that seem to have useful behavior (Machado et al. 2017, Figure 8 and 9). Deep covering options improves over their results as we only construct the options generated by the second eigenfunction which is likely to be more useful than the other ones.

---

### Official Review · AnonReviewer3 · 2019-10-23
**Official Blind Review #3**

**Rating:** 6

**Review:**

Summary
The authors introduce deep covering options, an online mechanism to extend the covering options to large state spaces. They claim their method discovers options that are task agnostic. The method is evaluated in sparse reward domains and claims to gain improvement in exploration and performance as well.  The authors extend the recent developments in eigenfunction estimation of the Laplacian to a principled approach for option discovery to non-linear function approximation.

Covering options compute the second smallest eigenvalue and the corresponding eigenvector f of the Laplacian exactly by solving a constrained optimization problem (Eq2). However, this requires the adjacency matrix A as input, and a constrained optimization problem is hard to solve using gradient-based methods. To overcome this, the authors propose an approximation of the computation of the Laplacian with Eq3. This allows them to have a constraint-free objective to compute the eigenfunction which is now dependent on the trajectories and avoids requiring the state-space graph. I believe the paper presents interesting ideas and is definitely a very useful contribution.  However, the paper needs work on thorough empirical analysis: could use more rigorous baselines especially to fairly evaluate the gains in exploration.

Detailed comments:
(I) Major Concern:  Figure 1 of this work is exactly the same as Figure 2 of the Jinnai et al., 2019b. It is not clear to me whether a) this is being referred for the purpose of giving intuitions. If yes, then it should be cited as Figure from Jinnai et al., 2019b or b) this is a new figure generated differently, it is not clear what the difference is. Overall, there is overlapping content and should be clarified what is original work and what is being referred to.

Reusing content from another paper without proper attribution is normally considered plagiarism. More precisely; Jinnai et al., 2019b. mention that the “second smallest eigenvalue of L is known as the algebraic connectivity of the graph and its corresponding eigenvector is called Fiedler vector” and caption this figure:
“Figure 2: The distance between the red state and all other states, measured via the Fiedler vector (left) and Euclidean distance (right). The Fiedler vector captures the connectivity of the graph, so distances measured using it reflect path lengths in the graph; the pair of nodes with the maximum and the minimum value are the farthest apart”
This  work (currently in review) captions this figure as:
“Figure 1: The distance between the red state and all other states, measured via the second eigenvector (left) and Euclidean distance (right). The second eigenvector captures the connectivity of the graph, so distances reflect path lengths in the graph; the pair of nodes with the maximum and minimum values are the farthest apart.” The only words changed are Fiedler vector to the second eigenvector. Please explain!

(II) An important step in the algorithm is line 3” identify an under-explored region in the state-space using the eigenfunctions”. Where does the parameter k come from? Is this a hand-designed parameter such as in PinBall it is somehow set to 30, for Mujoco tasks this is somehow set to 10, and for Atari, this is set to 4.  Would it be possible to comment on the hyperparameter threshold percentile k? Why is this a hard-coded choice? Would it be possible to learn this *simultaneously* as the options?

(III) Can you comment on to what extent do you see theoretical guarantees of covering options apply to the function approximation case as they no longer hold when going from tabular setting to the nonlinear function approximation? Would it be possible to also comment on what can be said about any guarantees at all if the state space is very very large? Forex: imagine a lifelong learning scenario where the environment is really big, and it is just not almost impossible to visit all states, how does this objective function of minimizing the upper bound on the expected cover time constitute the right choice?

(IV) Intuitively the pseudo-reward seems a lot related to goal-based rewards, where the skill is learned to reach different goals such as in DIAYN. Can you comment on how is this different and why is the proposed approach better in principle?
 Empirically, It is not clear why DIAYN was not compared as it is and with all the stated modifications.  In DIAYN, setting the initiation set to be the whole state space seems counterintuitive.

(V) Regarding the connectivity of the states to generate a diverse set of options: there seem to be connections to the work on constructing options using stronger guarantees Castro & Precup, 2011. It might be useful to comment on this and discuss this in the paper.

(VI) “We sampled 200 episodes of length 2000 with a uniform random policy to generate each option.” Would there be smarter ways to generate each option? Is there a reason to generate each option in this fashion?

(VII) Since the idea here is to connect states that are closer in terms of time, but further apart, how does the proposed approach comparison to successor options. I would imagine a comparison to successor options would be quite valuable to this. Please comment on this.

(VIII) “Termination set generated by deep covering options tends to be larger than..” Intuitively speaking, it is not clear why this is a good idea. Wouldn't we want the options to be peaked in places they terminate and initiate in, and therefore have smaller termination/initiation set?

(IX) In the Mujoco tasks: I would expect to see the baseline performance reported in the main paper i.e. DIAYN for continuous control tasks too. It is not clear by “did not outperform the baseline” as to how the proposed approach fairs in comparison to DIAYN. Looking into the appendix, It is still not clear why the plot of DIAYN in mujoco tasks is not included.
Since the key ideas in this work propose overcoming exploration as the main challenge, one would expect a comparison to the state-of-art in exploration, for instance [1]. Also since the work builds on eigenoptions, it would help compare with [2] as well. Without these comparisons, I find the empirical analysis rather weak.

(X) Pinball exploration visuals show concrete gains in state-space explored 4a-d. This is also evident in continuous control tasks 4e-f-g. However, in both cases, I find the comparison weak as the authors do not compare against state-of-the-art exploration baselines.

(XI)  The termination sets in ALE are interesting in that they do convey that options terminate in different regions of the state space: but one can also see options terminating in different regions of the state space in Harb et al, 2018 in Amidar here for example. It is also counter-intuitive as to why the options terminate in regions that do not overlap with visible goals for example in Montezuma Revenge in the key or skull. Perhaps one benefit here is that there is no reward information, but we do not see either the performance curve or the nature of options in ALE, so it is really hard to make a strong claim either way.

Although the options in pre-training were generated without a reward- I would recommend using these options in different tasks to make the claim “task agnostic options are discovered” much stronger.
[1] Count-based exploration with the successor representation.
[2] A laplacian framework for option discovery in reinforcement learning

Overall
Scales a principled approach to function approximation for deep covering options. The method seems to be computationally tractable. The approach can be applied to both settings where an unsupervised pre-training phase is available and also in a fully online setting.

However, DIAYN is the only baseline in Mujoco and is not shown. It’s unclear why other baselines were not used such as Eigenoptions, which seems like a very valid baseline. In addition, baselines of well-known exploration algorithms also have not been fully explored. The paper needs a more thorough evaluation of the proposed method to make the claims stronger.

Please note that although I have marked a weak reject, I am open to adjusting my score if the rebuttal addresses enough issues.

**Experience Assessment:**

I have published one or two papers in this area.

**Review Assessment: Checking Correctness Of Derivations And Theory:**

I assessed the sensibility of the derivations and theory.

**Review Assessment: Checking Correctness Of Experiments:**

I carefully checked the experiments.

**Review Assessment: Thoroughness In Paper Reading:**

I read the paper thoroughly.

---

> ### Author Response · Authors · 2019-11-12
> **Response to Reviewer #3**
>
> We appreciate the time you took reviewing our submission and hope that our response addresses some of your concerns.
>
>
> >  Figure 1 of this work is exactly the same as Figure 2 of the Jinnai et al., 2019b
>
> Thank you very much for pointing out. We added an acknowledgement to the Figure. “adapted from Jinnai et al. (2019b), Figure 2.” We apologize for the omission.
>
>
> >  Where does the parameter k come from? Is this a hand-designed parameter such as in PinBall it is somehow set to 30, for Mujoco tasks this is somehow set to 10, and for Atari, this is set to 4. Would it be possible to comment on the hyperparameter threshold percentile k?
>
> The threshold percentile is set smaller for tasks where the state-space size is large so that the termination set won’t be too large. Anecdotally, this parameter was quite easy to set by hand. Learning an appropriate setting for this hyperparameter automatically is future work.
>
>
> > Can you comment on to what extent do you see theoretical guarantees of covering options apply to the function approximation case as they no longer hold when going from tabular setting to the nonlinear function approximation?
>
> The advantage of having underlying theoretical guarantee is that we can hope that the method is more likely to have the desirable characteristic if we make the approximation finer grained. In fact, Wu et al. 2019 empirically showed that the eigenfunction constructed by their approximation method is fairly close to the exact solution in grid world.
>
>
> > It is not clear why DIAYN was not compared as it is and with all the stated modifications. In DIAYN, setting the initiation set to be the whole state space seems counterintuitive.
>
> Our code is indeed based on the implementation by the author of DIAYN (https://github.com/haarnoja/sac/blob/master/sac/algos/diayn.py) with several modifications to make it compatible to our implementation. We observed that the same set of hyperparameters do not outperform a non-hierarchical agent in our tasks. We suspect that the diversity objective matches nicely to tasks evaluated in Eysenbach et. al (2019) (e.g. Cheetah Hurdle, Ant Navigation) but is not as good a fit to our maze-like tasks.
>
>
> > “We sampled 200 episodes of length 2000 with a uniform random policy to generate each option.” Would there be smarter ways to generate each option?
>
> Our method is not restricted to samples from random trajectories but applicable to any valid trajectories. We chose to generate options from trajectories generated by a uniform random policy without using any task-specific information as our goal is to improve the performance of an agent which has no prior knowledge of the task.
>
>
> > how does the proposed approach comparison to successor options.
>
> The eigenfunctions of the Laplacian are equivalent to the eigenfunctions of the successor representation used by successor options (Stachenfeld et al. 2014). Therefore, both successor options and deep covering options are based on the same idea to identify subgoals in the state-space. The advantage of covering options is that it only uses the subgoal identified by the second eigenfunction which reduces the cover time the most among all the eigenfunctions in tabular domain whereas other eigenfunction methods use other eigenfunctions which reduces the cover time less than the second eigenfunction.
>
> Stachenfeld, Kimberly L., Matthew Botvinick, and Samuel J. Gershman. "Design principles of the hippocampal cognitive map." Advances in neural information processing systems. 2014.
>
>
> > “Termination set generated by deep covering options tends to be larger than..” Intuitively speaking, it is not clear why this is a good idea. Wouldn't we want the options to be peaked in places they terminate and initiate in, and therefore have smaller termination/initiation set?
>
> We do not claim that larger termination set is advantageous. We observed that deep covering options tend to have large termination set. How large the termination/initiation set should be is an open question.

---

> > ### Author Response · Authors · 2019-11-12
> > **(Cont.) Response to Reviewer #3**
> >
> > > I would expect to see the baseline performance reported in the main paper i.e. DIAYN for continuous control tasks too. It is not clear by “did not outperform the baseline”
> >
> > We added the learning curve of DIAYN for MuJoCo tasks in the paper. DIAYN did not outperform DDPG (with no options) in the tasks we evaluated.
> >
> >
> > > one would expect a comparison to the state-of-art in exploration, for instance [1]. Also since the work builds on eigenoptions, it would help compare with [2] as well. [1] Count-based exploration with the successor representation. [2] A laplacian framework for option discovery in reinforcement learning
> >
> > [1]: The goal of deep covering options is to construct a skill to efficiently explore the state-space agnostic to its reward structure. Therefore, generated options can be used for any tasks in the same state dynamics, thus transferable across tasks which is hard to achieve by non-option based methods. In fact, our method is orthogonal to non-option based exploration methods, so we can apply both algorithms.
> > [2]: Our method is an extension of covering options that is shown to outperform eigenoptions in tabular domains (Jinnai et al 2019b).

---

> > > ### Comment · AnonReviewer3 · 2019-11-14
> > > **Follow up comments**
> > >
> > > Thank you for the clarification on the comments.
> > >
> > > >The threshold percentile is set smaller for tasks where the state-space size is large so that the termination set won’t be too large. >We observed that deep covering options tend to have large termination set.
> > >  If we chose k such that for large state-spaces the termination sets won't be too large, yet we observe that deep covering options tend to have large termination set. Could you elaborate on the impact of different values of k. It would be nice to have an ablation study with different values of k.
> > >
> > > >We added the learning curve of DIAYN for MuJoCo tasks in the paper. DIAYN did not outperform DDPG (with no options) in the tasks we evaluated.
> > > Thank you for adding these results.
> > >
> > > - The termination sets in ALE are interesting in that they do convey that options terminate in different regions of the state space: but one can also see options terminating in different regions of the state space in Harb et al, 2018 in Amidar here for example. It is also counter-intuitive as to why the options terminate in regions that do not overlap with visible goals for example in Montezuma Revenge in the key or skull.  We do not see either the performance curve or the nature of options in ALE, so it is really hard to make a strong claim either way in ALE.
> > > Any comments on this would be great.
> > >
> > > >Finding efficient options to improve the exploration in the ATARI domain is very challenging. While Machado et al. (2017; 2018) showed that eigenoptions is applicable to the ATARI domain, they did not present learning curves.
> > > But we do see learning curves for option-critic methods [1,2] for instance in ALE. Although I agree that the options found in [1,2] do not aim to improve exploration, it would still be meaningful to evaluate performance curves of deep-covering options in ALE, and how they compare to other existing methods.
> > >
> > > [1] Bacon, Pierre-Luc, Jean Harb, and Doina Precup. "The option-critic architecture." Thirty-First AAAI Conference on Artificial Intelligence. 2017.
> > > [2] Tiwari, Saket, and Philip S. Thomas. "Natural option critic." Proceedings of the AAAI Conference on Artificial Intelligence. Vol. 33. 2019.

---

> > > > ### Author Response · Authors · 2019-11-15
> > > > **Response to the follow up comments**
> > > >
> > > > Thank you very much for the comments.
> > > >
> > > > > If we chose k such that for large state-spaces the termination sets won't be too large, yet we observe that deep covering options tend to have large termination set. Could you elaborate on the impact of different values of k. It would be nice to have an ablation study with different values of k.
> > > >
> > > > We will add an ablation study with different values of k to understand its impact to the performance in the camera ready.
> > > > We didn’t have enough resources to run the ablation study before the deadline of the rebuttal.
> > > >
> > > >
> > > > > The termination sets in ALE are interesting in that they do convey that options terminate in different regions of the state space: but one can also see options terminating in different regions of the state space in Harb et al, 2018 in Amidar here for example. It is also counter-intuitive as to why the options terminate in regions that do not overlap with visible goals for example in Montezuma Revenge in the key or skull.  We do not see either the performance curve or the nature of options in ALE, so it is really hard to make a strong claim either way in ALE.
> > > > > But we do see learning curves for option-critic methods [1,2] for instance in ALE. Although I agree that the options found in [1,2] do not aim to improve exploration, it would still be meaningful to evaluate performance curves of deep-covering options in ALE, and how they compare to other existing methods.
> > > >
> > > > The goal of deep covering options is to construct a skill to efficiently explore the state-space agnostic to its reward structure whereas option-critic methods learn options using the reward from the environment. Thus, our method is orthogonal to option-critic so we can apply both algorithms to construct options. One may initialize options using deep covering options and then refine them by option-critic using the reward information.

---

> > > > > ### Comment · AnonReviewer3 · 2019-11-15
> > > > > **Updating my score**
> > > > >
> > > > > > both successor options and deep covering options are based on the same idea to identify subgoals in the state-space.
> > > > > This recent work on successor options [1] might be relevant to add to the discussion as well.
> > > > >
> > > > > Based on the authors' clarifications and more reading: considering it is really challenging to extend the theoretical ideas in Jinnai et al 2019b to the function approximation settings as also evident in [1] for successor options, I am happy to update my score.  I think the ideas proposed are a valuable contribution to the field.
> > > > >
> > > > > -Please add ablation studies for different values of k.
> > > > >
> > > > > -I  recommend adding/editing the paper to include all clarifications addressed during the rebuttal.
> > > > >
> > > > > -I  also recommend that the authors add a clear discussion on why ALE is hard for the proposed method, so much that no learning curves have been shown. It would be great if the authors can also add the learning curves even if in the appendix. It is okay to not have outperforming curves in all domains! Especially since the contribution in point is to show how Jinnai et al 2019b can be extended to the function approximation setting.
> > > > >
> > > > > [1] Ramesh, Rahul, Manan Tomar, and Balaraman Ravindran. "Successor Options: An Option Discovery Framework for Reinforcement Learning." arXiv preprint arXiv:1905.05731 (2019).

---

### Official Review · AnonReviewer2 · 2019-10-25
**Official Blind Review #2**

**Rating:** 6

**Review:**

The paper proposes an algorithm to extend the recently proposed method of “covering options” from a tabular setting to continuous state spaces (or large discrete state spaces). The proposed algorithm approximately computes the second eigenfunction of the normalized laplacian of the state space, uses it to identify an under-explored region and trains an option to terminate in such a region. Each new learnt option is added to the initial set of primitive actions and a policy over this growing set of actions is learnt separately. An online algorithm is also proposed that does the above option learning process intermittently in addition to training for an external task. The paper shows empirical evidence of better or equal performance to base algorithms which do not discover options, prior work such as DIAYN (Eysenbach et. al, 2019) (that discover options via mutual information maximization between visited states and options), as well as ablations of their proposed method with different number of options.

I vote for weak reject due to (1) the idea of covering options (Jinnai et. al., 2019b) and the approximation for the graph laplacian (Wu et. al., 2019) both have been shown in prior work and the novelty in this paper seems to be limited to putting together these two ideas, and (2) the paper shows quantitative results for options discovered for simple environments whereas only qualitative options are shown for harder exploration tasks, and (3) given that exploration is a key problem being addressed, a comparison to other exploration algorithms which are non-option based methods has not been shown -- which makes for a weak argument for using an option-based method for exploration as opposed to existing methods for exploration.

Other comments:
- The paper does a good job at explaining covering options, the approximations involved in their algorithm and how options can be learnt with the help of such approximations. It makes sense that an algorithm that takes into account state connectivity will do better than DIAYN (Eysenbach et. al., 2019) which just promotes diversity of visited states across options.

- The paper made some assumptions on the implementation of DIAYN, citing lack of details in Eysenbach et. al (2019). Were these details not available in the code released by DIAYN on their project page? (I’m not sure if I should post the link here, but it is easy to find). I have some concern that a fair comparison may not have been made due to the paper’s implementation of this baseline, but I am also not sure how important of a comparison this is given that their method for discovering options is quite different. I would argue that DIAYN is different enough that the proposed method deserves comparisons with methods for exploration that do not use options.

- In the intro, “...extend a theoretically principled approach for option discovery to the non-linear function approximation case” seems to be highly misleading, since it suggests that the theoretical guarantees are extended, which is not the case in this paper as the approximations have no states guarantees.

The proposed method seems to have the potential to be a good at exploration but the paper does not show quantitative experiments for hard exploration tasks such as Montezuma’s Revenge, or other Atari ALE environments. I am curious to see how many levels of Montezuma’s revenge can be covered by simply applying deep covering options to it.

References:
All references are same as the paper.

After rebuttal: The authors have addressed most major concerns and I am increasing my score to weak accept.


**Experience Assessment:**

I have read many papers in this area.

**Review Assessment: Checking Correctness Of Derivations And Theory:**

N/A

**Review Assessment: Checking Correctness Of Experiments:**

I carefully checked the experiments.

**Review Assessment: Thoroughness In Paper Reading:**

I read the paper at least twice and used my best judgement in assessing the paper.

---

> ### Author Response · Authors · 2019-11-12
> **Response to Reviewer #2**
>
> We appreciate the time you took reviewing our submission and hope that our response addresses some of your concerns.
>
>
> > the idea of covering options (Jinnai et. al., 2019b) and the approximation for the graph laplacian (Wu et. al., 2019) both have been shown in prior work and the novelty in this paper seems to be limited to putting together these two ideas
>
> One of the main limitations of eigenoptions is that the existing algorithms discover too many options at once and it is not clear how to prune them. The work on covering options addressed this limitation but only in a constrained, theoretical setting. Extending this idea to the function approximation setting is actually really challenging. We argue that the strength of our paper is to show that covering options can be easily extensible to function approximator setting by applying and adapting the techniques shown by Wu et al.
>
>
> > only qualitative options are shown for harder exploration tasks
>
> Finding efficient options to improve the exploration in the ATARI domain is a very challenging task. In fact, Machado et al. (2017; 2018) showed that eigenoptions is applicable to the ATARI domain but did not present learning curves. They constructed 1024 options from all the generated eigenfunctions and manually picked some of the options out of them that seem to have useful behavior (Machado et al. 2017, Figure 8 and 9). Deep covering options improves over their results as we only construct the options generated by the second eigenfunction which is likely to be more useful than the other ones.
>
>
> > a comparison to other exploration algorithms which are non-option based methods has not been shown
>
> The goal of deep covering options is to construct a skill to efficiently explore the state-space agnostic to its reward structure. Therefore, generated options can be used for any tasks in the same state dynamics, thus transferable across tasks which is hard to achieve by non-option based methods. In fact, our method is orthogonal to non-option based exploration methods, so we can apply both of the algorithms.
>
>
> > In the intro, “...extend a theoretically principled approach for option discovery to the non-linear function approximation case” seems to be highly misleading, since it suggests that the theoretical guarantees are extended, which is not the case in this paper as the approximations have no states guarantees.
>
> Thank you very much for pointing out. We changed the sentence to “...extend a principled approach for option discovery to the non-linear function approximation case”.
>
>
> > The paper made some assumptions on the implementation of DIAYN, citing lack of details in Eysenbach et. al (2019). Were these details not available in the code released by DIAYN on their project page? (I’m not sure if I should post the link here, but it is easy to find)
>
> Our code is indeed based on the implementation by the author of DIAYN (https://github.com/haarnoja/sac/blob/master/sac/algos/diayn.py) with several modifications to make it compatible to our implementation. We observed that the same set of hyperparameters do not outperform a non-hierarchical agent in our tasks. We suspect that the diversity objective matches nicely to tasks evaluated in Eysenbach et. al (2019) (e.g. Cheetah Hurdle, Ant Navigation) but is not as good a fit to our maze-like tasks.

---

> > ### Author Response · Authors · 2019-11-15
> > **Response to Reviewer #2**
> >
> > Please let us know if our response addresses your concerns, or if some clarification is needed.

---

### Decision · Program_Chairs · 2019-12-19

**Decision:**

Accept (Poster)

**Comment:**

This paper considers options discovery in hierarchical reinforcement learning. It extends the idea of covering options, using the Laplacian of the state space discover a set of options that reduce the upper bound of the environment's cover time, to continuous and large state spaces. An online method is also included, and evaluated on several domains.

The reviewers had major questions on a number of aspects of the paper, including around the novelty of the work which seemed limited, the quantitative results in the ATARI environments, and problems with comparisons to other exploration methods. These were all appropriately dealt with in the rebuttals, leaving this paper worthy of acceptance.